# Ume6-dependent pathways of morphogenesis and biofilm formation in *Candida auris*

Marine Louvet,[1] Jizhou Li,[1] Danielle Brandalise,[1] Daniel Bachmann,[1] Francisco Sala de Oyanguren,[2] Danny Labes,[2] Nicolas Jacquier,[1] Christel Genoud,[3] Antonio Mucciolo,[3] Alix T. Coste,[1] Dominique Sanglard,[1] Frederic Lamoth[1,4]

**ABSTRACT** *Candida auris* is a yeast pathogen causing nosocomial outbreaks of candidemia. Its ability to adhere to inert surfaces and to be transmitted from one patient to another via medical devices is of particular concern. Like other *Candida* spp., *C. auris* has the ability to transition from the yeast form to pseudohyphae and to build biofilms. Moreover, some isolates have a unique capacity to form aggregates. These morphogenetic changes may impact virulence. In this study, we demonstrated the role of the transcription factor Ume6 in *C. auris* morphogenesis. Genetic hyperactivation of Ume6 induced filamentation and aggregation. The Ume6-hyperactivated strain (*UME6*^HA) also exhibited increased adhesion to inert surface and formed biofilms of higher biomass compared to the parental strain. Transcriptomic analyses of *UME6*^HA revealed enrichment of genes encoding for adhesins, proteins involved in cell wall organization, sterol biosynthesis, and aspartic protease activities. The three most upregulated genes compared to wild-type were those encoding for the agglutin-like sequence adhesin Als4498, the *C. auris*-specific adhesin Scf1, and the hypha-specific G1 cyclin-related protein Hgc1. The deletion of these genes in the *UME6*^HA background showed that Ume6 controls filamentation via Hgc1 and aggregation via Als4498 and Scf1. Adhesion to inert surface was essentially triggered by Scf1. However, Als4498 and Hgc1 were also crucial for biofilm formation. Our data show that Ume6 is a universal regulator of *C. auris* morphogenesis via distinct modulators.

**IMPORTANCE** *C. auris* represents a public health threat because of its ability to cause difficult-to-treat infections and hospital outbreaks. The morphogenetic plasticity of *C. auris*, including its ability to filament, to form aggregates or biofilms on inert surfaces, is important to the fungus for interhuman transmission, skin or catheter colonization, tissue invasion, antifungal resistance, and escape of the host immune system. This work deciphered the importance of Ume6 in the control of distinct pathways involved in filamentation, aggregation, adhesion, and biofilm formation of *C. auris*. A better understanding of the mechanisms of *C. auris* morphogenesis may help identify novel antifungal targets.

**KEYWORDS** aggregation, pseudohyphae, biofilm, adhesin, transcriptomic

Address correspondence to Frederic Lamoth, Frederic.Lamoth@chuv.ch.

Marine Louvet and Jizhou Li contributed equally to this article. Author order was determined in order of increasing seniority.

The authors declare no conflict of interest.

See the funding table on p. 15.

*C*andida auris (recently renamed *Candidozyma auris*) is a pathogenic yeast, which has attracted a lot of attention during the last decade because of its ability to develop resistance to antifungal drugs and to cause nosocomial outbreaks (1–4). Investigations of *C. auris* outbreaks revealed the presence of the fungus on inert surfaces, such as the reusable medical instruments or the room furniture (5–7). In addition to being resistant to many standard surface disinfectants, *C. auris* can survive and persist on dry or moist inert surfaces including plastic, steel, or wood (8–10). Similar to other *Candida* spp., *C. auris* can develop biofilms on host or inert surfaces (11–13). Biofilms are extracellular

matrices that protect the cells from the host immune system and the antifungal drugs (14). The ability of yeasts to form biofilms may contribute to virulence via colonization of medical devices, such as intravascular catheters (14). Biofilm-forming ability of *Candida* isolates has been associated with bad prognosis among patients with *Candida* bloodstream infections (15, 16).

Other morphogenetic changes of *C. auris*, such as filamentation and aggregation, may contribute to virulence. *C. auris* was initially thought to not form hyphae or pseudohyphae (17). However, filamentation has been observed in specific culture media or under particular stress conditions (18, 19). Filamentous forms resembling true hyphae of *Candida albicans* have been observed in *C. auris* following passage through the organs of mammalian bodies (20). While the yeast form may favor adhesion to host cells or dissemination in blood, the transition to pseudohyphae may contribute to tissue invasion, escape to the immune system, and biofilm formation (21). Cell aggregation is a specific feature of *C. auris* consisting of clusters of cells remaining attached to each other, which results from a defect in cell division and/or increased inter-cell adhesion (17). Aggregation was found to be clade-dependent, being commonly observed among isolates of clade III (18). Aggregating isolates were found to be less virulent than the non-aggregating ones in invertebrate models of infection (17, 22). Among *C. auris* clinical isolates, non-aggregative variants are more prone to cause candidemia, while the aggregative variants are more frequently associated with colonization with a higher propensity to form biofilms (23).

Transcriptional analyses have identified genes that may be involved in aggregation, filamentation, and biofilm formation of *C. auris*, including genes encoding for adhesins and metabolic processes (12, 20, 24–27). However, transcription factors regulating morphogenesis are still unexplored in *C. auris*. Because the zinc cluster transcription factor Ume6 is known to be a key regulator of the switch from yeast to hyphal form and of biofilm formation in *C. albicans* (28–31), we explored its role in *C. auris*.

## RESULTS

### Ume6 is involved in aggregation and pseudo-filamentation of *C. auris*

To assess the role of *UME6* in *C. auris*, we generated from the IV.1 background strain a mutant with deletion of *UME6* (*ume6Δ*) and a mutant in which Ume6 was hyperactivated. Hyperactivation of Ume6 was achieved by substitution of the native *UME6* promoter by the promoter of the *ADH1* gene ($P_{ADH1}$) for constitutive overexpression and by the addition of a 3 HA tag (*UME6^HA*) to its C-terminus (Table S1). The addition of a 3 HA tag to the C-terminal domain of zinc cluster transcription factors was shown to be efficient for their artificial activation in *C. albicans* and *C. auris*, presumably by relieving an activation domain from repression by an inhibitory domain (32–35).

No growth defect or phenotypic alteration was observed in the *ume6Δ* strain (Fig. S1A). Despite the absence of growth defect, the *UME6^HA* strain displayed an altered morphological aspect of the colony with irregular and wrinkled borders (Fig. 1A). Differential interference contrast (DIC) microscopy and scanning electron microscopy (SEM) revealed the presence of aggregates (i.e., clusters of yeast cells without separation after budding) and formation of pseudohyphae in *UME6^HA*, which was not the case in IV.1 (Fig. 1B and C). These observations support the role of Ume6 in the aggregation and pseudo-filamentation of *C. auris*.

To further analyze the role of Ume6 in pseudofilamentation, we performed image flow cytometry assays of the *UME6^HA* and IV.1 strains. The morphological aspect of the cells was analyzed according to their horizontal/vertical length ratio and defined as "non-budding," "budding," or "pseudohyphae" for ratios of 0.8–1, 0.5–0.8, and <0.5, respectively. The proportions of non-budding cells, budding cells, and pseudohyphae were 52.1%, 47.0%, and 0.9% for IV.1 and 17.1%, 56.9%, and 26.0% for *UME6^HA*, respectively (Fig. 2). Measurement of cell length (i.e., maximal diameter) by image flow cytometry showed a higher proportion of cells measuring between 5–10 µm, 10–15 µm, or >15 µm in *UME6^HA* compared to IV.1 (Fig. S2).

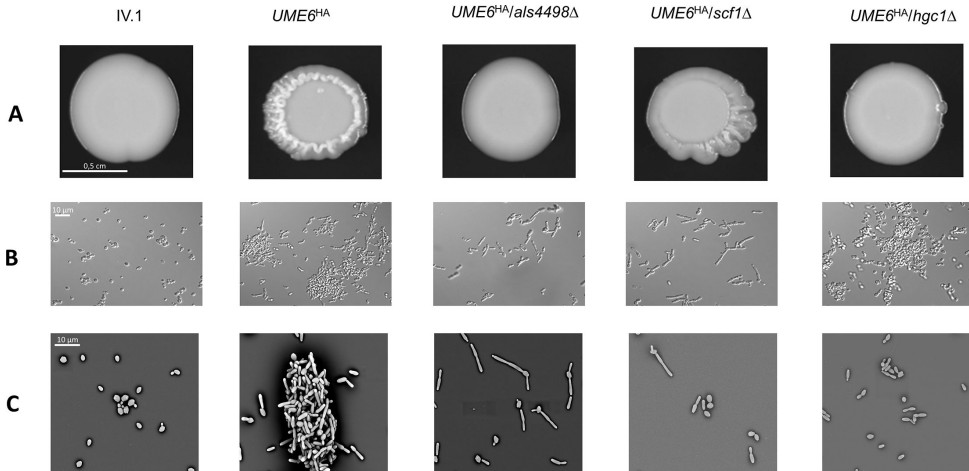

**FIG 1** Phenotypic aspects of the *Candida auris* strains used in this study. (A) Macroscopic pictures of the colonies after overnight culture on solid yeast extract-peptone-dextrose (YEPD) at 37°C. (B) Differential interference contrast (DIC) microscopy after overnight growth in liquid YEPD at 37°C. The images were captured at magnification 630× with a ZEISS Axiocam 305 (Carl Zeiss AG, Oberkochen, Germany) using Zen software. (C) Pictures taken by SEM at magnification 2,400×. Size standards are shown in the first column of each panel.

## Transcriptional profile resulting from Ume6 hyperactivation

As a next step, we performed transcriptomic analyses (RNA sequencing) of *UME6*^HA and IV.1 to identify the genes that are upregulated and downregulated following Ume6 hyperactivation. This analysis confirmed the successful upregulation of *UME6* (SBP28_002473/B9J08_000592) in *UME6*^HA (2,210-fold compared to IV.1) (File S1). *UME6*^HA exhibited significantly higher expression (i.e., ≥2-fold increase and *P*-value ≤ 0.05 compared to IV.1) and lower expression (i.e., ≥ 2-fold decrease and *P*-value ≤ 0.05 compared to IV.1) of 384 and 88 genes, respectively (File S1). Gene Ontology analysis (File S2) revealed an enrichment in gene products located to cell wall and involved in cell wall organization (e.g., adhesins), sterol biosynthesis, and aspartic protease activities (Fig. 3A). GO term enrichment of downregulated genes was principally involved in ribosome biogenesis (Fig. 3B). These results suggest that Ume6 controls processes regulating growth and proliferation.

We next compared the transcriptional profile of *UME6*^HA with other published data sets (*n* = 90) using gene set enrichment analysis (GSEA). As shown in Fig. S3, the resulting network of data sets showed significant overlap with RNAseq profiles of genes upregulated by the *in vivo* yeast to filament transition [node "FILAMENT VS YEAST_UP" (20)]. The enrichment plot based on this node (Fig. 4) showed that 46 genes (listed in File S3) were commonly upregulated between yeast to filament transition and *UME6*^HA. Together with other data sets containing genes upregulated in contact with blood cells [node "BLOOD_60 MIN_UP" (36)] and genes upregulated by deprivation of Hsp90 [node "HSP90_TET_UP", (37)], these conditions have in common that they favor the filamentous phase of *C. auris*, which is consistent with the morphological effect of *UME6*^HA on *C. auris*.

Based on these results, we decided to further investigate the link between Ume6 and the three genes exhibiting the highest overexpression in *UME6*^HA (>100-fold change): the agglutinin-like sequence (ALS) adhesin SBP28_004635/B9J08_004498 (further referred as *ALS4498* according to the nomenclature proposed by Santana et al.) (27), the *C. auris*-specific adhesin SBP28_003606/B9J08_001458 (surface colonization factor 1, further referred as *SCF1*) (27), and the hypha-specific G1 cyclin-related protein SBP28_004253/B9J08_004946 (ortholog of *C. albicans HGC1*, further referred as *HGC1*) (38, 39). For this purpose, we deleted these genes in IV.1 and *UME6*^HA to generate the *als4498Δ*, *scf1Δ*, *hgc1Δ*, *UME6*^HA/*als4498Δ*, *UME6*^HA/*scf1Δ*, and *UME6*^HA/*hgc1Δ* strains, respectively (Table S1).

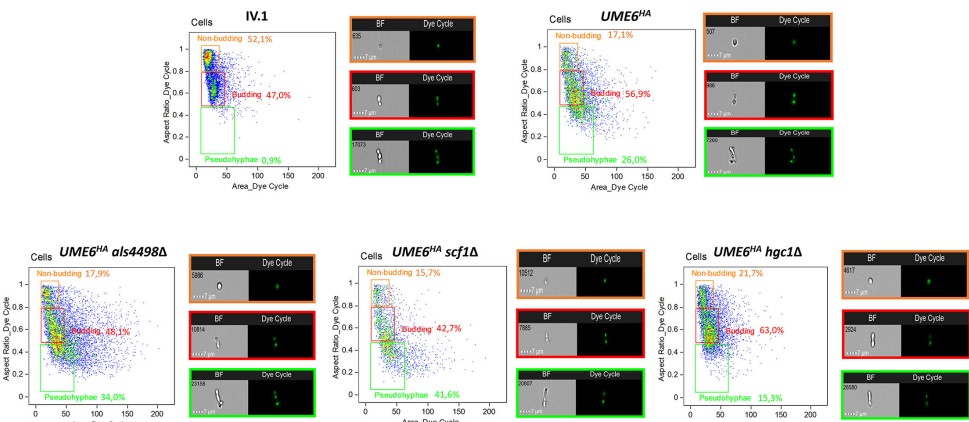

**FIG 2** Analysis of pseudohyphae formation by image flow cytometry. Graphical representation of the *C. auris* strains used in this study according to their morphological aspect. Yeast cells were separated from debris by the exclusion of low-frequency events (open squares) and classified according to their aspect ratio (i.e., major/minor axis ratio, *y*-axis) as "non-budding" (ratio 0.8–1, orange square), "budding" (ratio 0.5–0.8, red square), or "pseudohyphae (ratio <0.5, green square). The proportions of the different cell subpopulations (non-budding, budding, pseudohyphae) are expressed in percentages. Representative pictures in bright-field (BF) and Green Dye-Cycle (staining cell nuclei) captions are provided for each subpopulation: non budding (orange square), budding (red square), pseudohyphae (green square).

## Ume6 controls aggregation via Als4498 and Scf1 and pseudohyphae formation via Hgc1

The deletion of *ALS4498*, *SCF1,* and *HGC1* in the IV.1 strain did not result in any growth defect or morphological alteration (Fig. S4). However, distinct effects were observed following the deletion of these genes in *UME6*$^{HA}$. The morphological aspect of the *UME6*$^{HA}$ colony with irregular and wrinkled borders on solid agar was conserved in *UME6*$^{HA}$/*scf1*Δ but not in *UME6*$^{HA}$/*als4498*Δ and *UME6*$^{HA}$/*hgc1*Δ (Fig. 1A). By DIC microscopy and SEM, the aggregation observed in *UME6*$^{HA}$ was still present in *UME6*$^{HA}$/*hgc1*Δ but not in *UME6*$^{HA}$/*als4498*Δ and *UME6*$^{HA}$/*scf1*Δ (Fig. 1B and C). The ability of *UME6*$^{HA}$ to form pseudohyphae was conserved in *UME6*$^{HA}$/*als4498*Δ and *UME6*$^{HA}$/*scf1*Δ, while it was abolished in *UME6*$^{HA}$/*hgc1*Δ (Fig. 1B and C).

The morphological aspect of the mutants was analyzed by imaging flow cytometry assay (as described above). Compared to *UME6*$^{HA}$, the proportion of pseudohyphae was higher in *UME6*$^{HA}$/*scf1*Δ (41.6% vs 26.0%) and *UME6*$^{HA}$/*als4498*Δ (34.0% vs 26.0%), while it was lower in *UME6*$^{HA}$/*hgc1*Δ (15.3% vs 26.0%) (Fig. 2). When compared to *UME6*$^{HA}$,

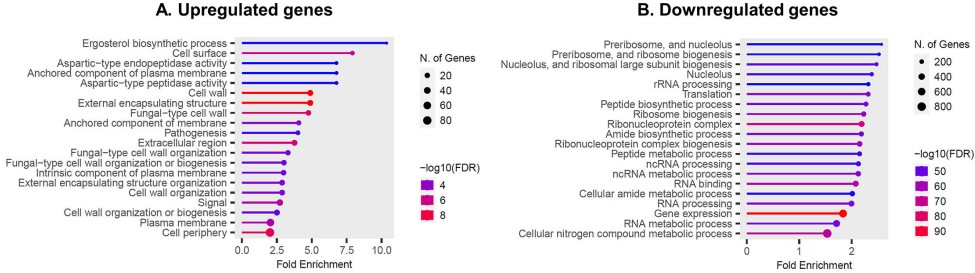

**FIG 3** Gene ontology (GO) analysis of upregulated genes (A) and downregulated genes (B). Examples of regulated genes belonging to specific enriched GO terms are given below: Cell surface and cell wall organization: SBP28_003648/B9J08_001507:GAS4; SBP28_004635/B9J08_004498:Als-like gene; SBP28_004797/B9J08_004410:CRH1; SBP28_000823/B9J08_003251:EXG1. Sterol biosynthesis: SBP28_001236/B9J08_002817:ERG28; SBP28_002155/B9J08_000261:ERG1; SBP28_001049/B9J08_003026:ERG24; SBP28_003595/B9J08_001448: RG11; SBP28_004192/B9J08_005007:ERG7. Aspartic protease activities: SBP28_003137/B9J08_002149: YPS7; SBP28_002287/B9J08_000398; YPS3.

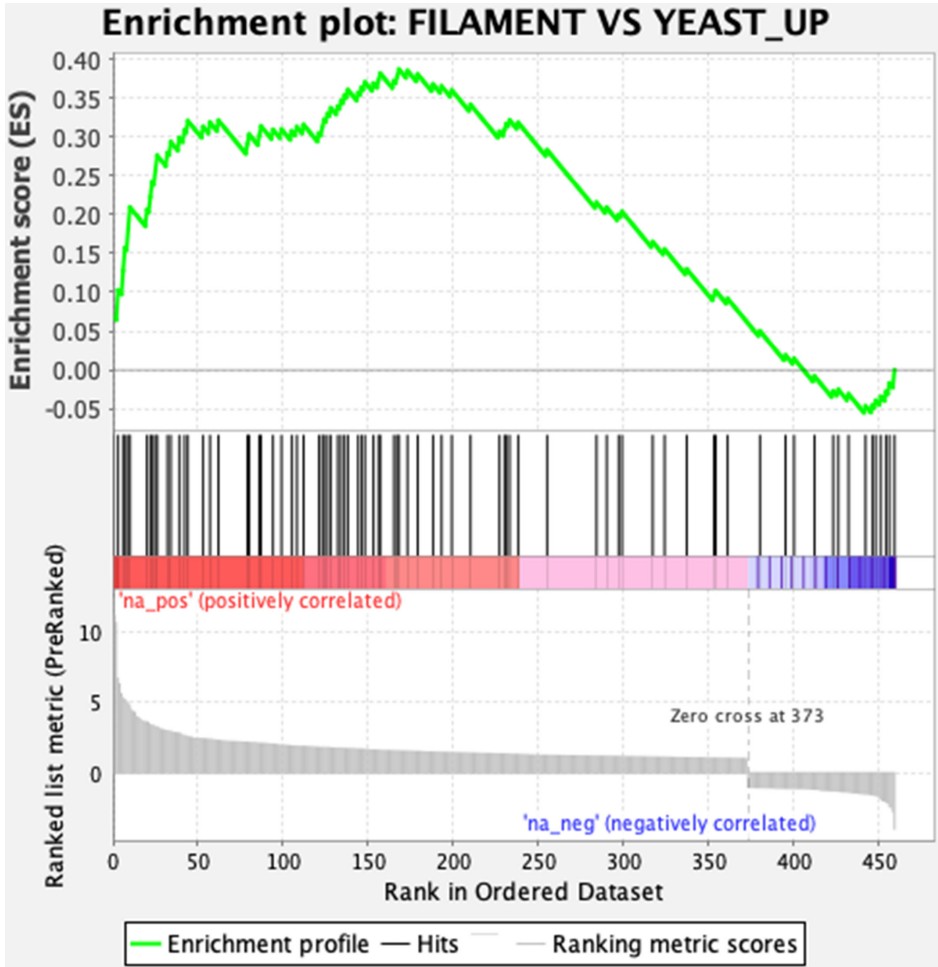

**FIG 4** Gene set enrichment plot between genes upregulated by *UME6*[HA] and genes upregulated by the yeast to filamentous phase (20). The data were obtained by extracting data from the GSEA Report for Data set Ume6.rnk (File S1) for FILAMENT VS YEAST_UP and show the profile of the running enrichment score (ES) and positions of GeneSet members on the rank-ordered list (Ume6.rnk).

*UME6*[HA]/*als4498*Δ, and *UME6*[HA]/*scf1*Δ exhibited a higher proportion of longer pseudohyphae (>10 µm), while this proportion was lower in *UME6*[HA]/*hgc1*Δ (Fig. S2).

These results suggest that Ume6 operates via distinct pathways in the control of morphogenesis in *C. auris*. While its impact on aggregation is mediated via Als4498 and Scf1, its role in pseudofilamentation is mediated via Hgc1.

## Ume6 controls adhesion to inert surface via Scf1, but Als4498 and Hgc1 are also crucial for biofilm formation

As Ume6 is known to be involved in adhesion and biofilm formation in *C. albicans* (28), we investigated the link between Ume6 and its downstream regulated proteins Als4498, Scf1 and Hgc1 in these processes in *C. auris*.

Adhesion to fluorescent polystyrene microspheres as a readout for adhesion to abiotic surfaces was analyzed by flow cytometry . No difference in adhesion percentage was observed between *ume6*Δ and IV.1 (Fig. S1B). However, *UME6*[HA] exhibited a significantly higher percentage of adhesion to microspheres compared to IV.1 (67.7% vs 24.8%, respectively, $P = 0.0001$) (Fig. 5). Compared to *UME6*[HA], adhesion was significantly decreased in *UME6*[HA]/*scf1*Δ and slightly increased in *UME6*[HA]/*als4498*Δ, while it was similar in *UME6*[HA]/*hgc1*Δ (Fig. 5). These observations were confirmed by imaging flow cytometry

(Fig. S5 and S6). These data confirm the crucial role of Scf1 in adhesion to inert surfaces as recently described (27, 40).

Biofilm formation on polystyrene surface was quantified by a crystal violet assay. Biofilm formation was similar between *ume6Δ* and IV1 (Fig. S1C). However, the biomass of adherent cells was significantly higher in *UME6^HA* compared to IV.1 (Fig. 6). The deletion of *ALS4498*, *SCF1,* and *HGC1* in *UME6^HA* all resulted in a significant decrease in biofilm formation, which was more pronounced in *UME6^HA/als4498Δ* (Fig. 6).

Analysis by confocal microscopy showed that *UME6^HA* produced a denser and thicker biofilm compared to IV.1 (Fig. 7). However, the density and thickness of the biofilm was significantly decreased in *UME6^HA/als4498Δ*, *UME6^HA/scf1Δ*, and *UME6^HA/hgc1Δ* compared to *UME6^HA* (Fig. 7).

These results show that Ume6 triggers biofilm formation of *C. auris* via different mechanisms involving all three downstream effectors (Als4498, Scf1, Hgc1).

## Ume6 hyperactivation increases antifungal resistance in biofilm conditions

Antifungal susceptibility testing was performed for IV.1 and *UME6^HA* in both planktonic and biofilm conditions. According to the Clinical and Laboratory Standards Institute (CLSI) protocol (41), minimal inhibitory concentrations (MIC) of fluconazole, amphotericin B, and micafungin were similar for IV.1 and *UME6^HA* (4 µg/mL, 2 µg/mL, and 0.25 µg/mL, respectively). In biofilm conditions, *UME6^HA* exhibited similar minimal biofilm eradication concentration ($MBEC_{50}$) to fluconazole compared to IV.1 (16 µg/mL), but higher MBEC to amphotericin B (32–64 µg/mL vs 8–16 µg/mL) and micafungin (16–32 µg/mL vs

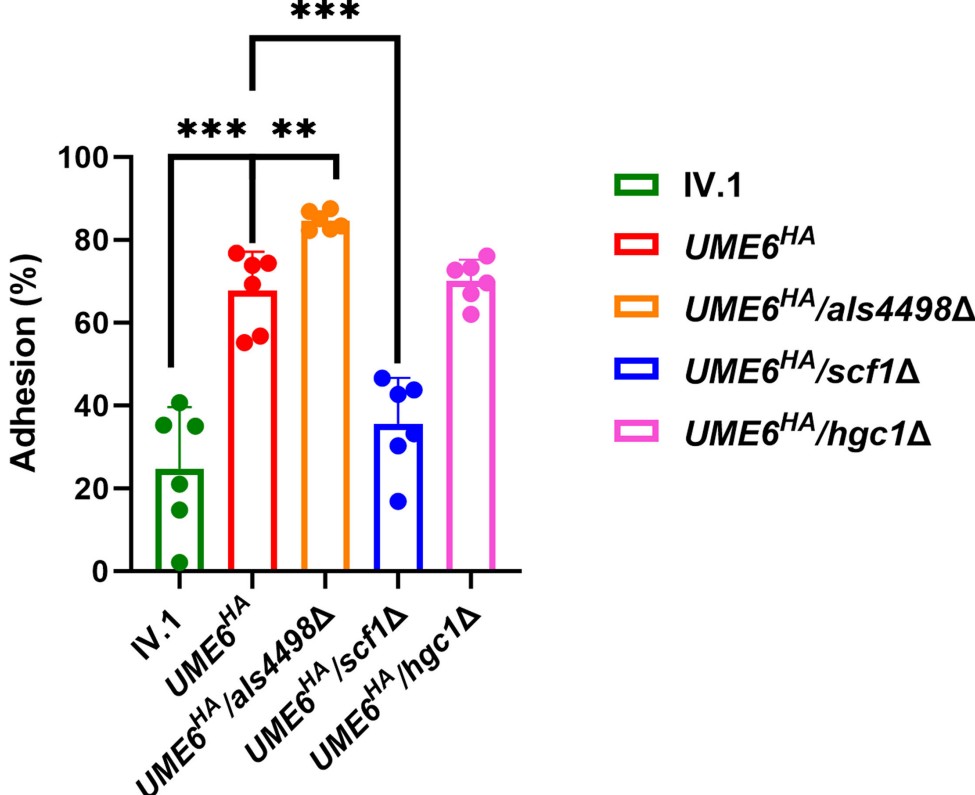

**FIG 5** Adhesion assay by flow cytometry. Percentage of adhesion of yeast cells to fluorescent polystyrene microspheres (*y*-axis) for the different strains (*x*-axis). Results are expressed as means with standard deviations (error bars) of technical triplicates and biological duplicates. Cells were delineated by uptake of DyeCycle Green (emission: $\lambda_{em} \approx 488$ nm). Following initial gating on DyeCycle Green-positive events, cells were further analyzed for adherence to fluorescent FluoSpheres carboxylate-modified microspheres (emission: $\lambda_{em} \approx 645$ nm). Statistical analysis was performed using unpaired *t*-test with significant *P*-value defined as ≤0.05 (* ≤0.05, ** ≤0.01, *** ≤0.001, **** ≤0.0001).

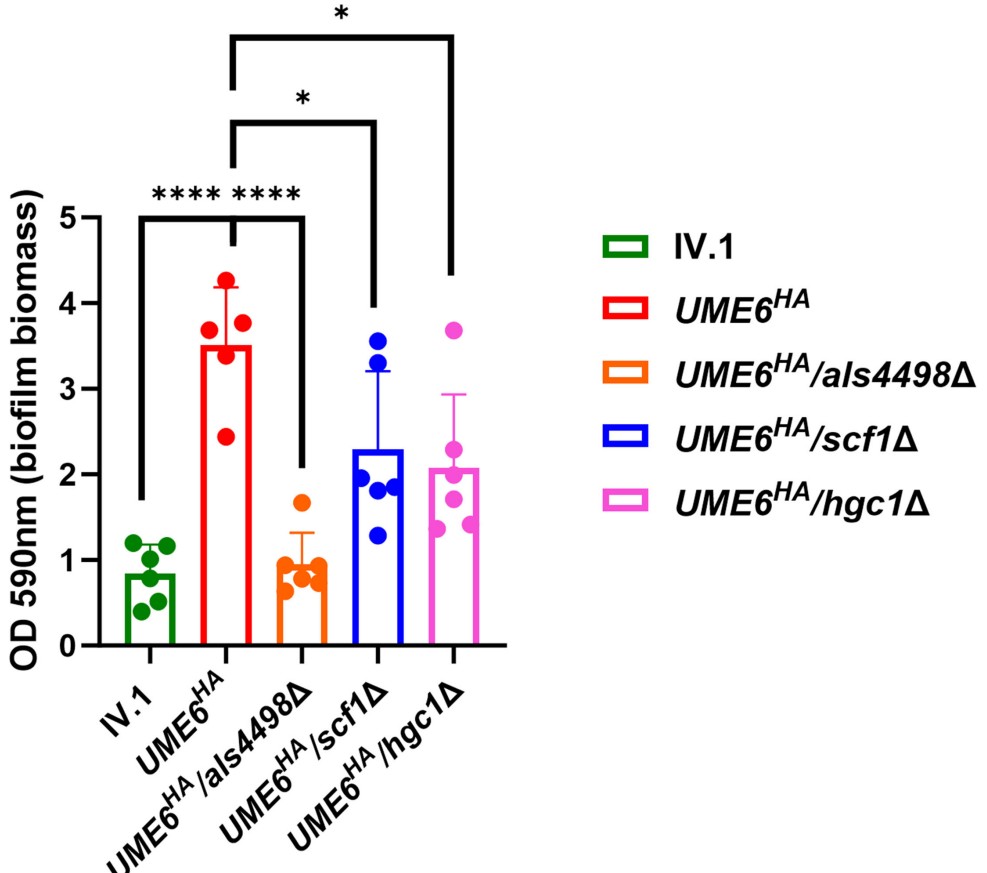

**FIG 6** Quantification of biofilm formation by crystal violet assay. Absorbance, which is representative of the biomass of adherent cells to the polystyrene surface, was measured at 590 nm after 24 h incubation (*y*-axis) for the different strains (*x*-axis). Results are expressed as means with standard deviations (error bars) of technical triplicates and biological duplicates. Statistical analysis was performed using unpaired *t*-test with significant *P*-value defined as ≤0.05 (* ≤0.05, ** ≤0.01, *** ≤0.001, **** ≤0.0001).

0.25–0.5 µg/mL). These results show that the biofilm resulting from Ume6 hyperactivation is resistant to biofilm-active antifungal drugs (e.g., amphotericin B, micafungin), in particular to micafungin.

### Ume6 is not essential for virulence in a *Galleria* model of invasive candidiasis

Finally, we assessed the role of Ume6 in *C. auris* virulence. Groups of larvae of *Galleria mellonella* were infected with the IV.1, *UME6*$^{HA}$, or *ume6*Δ strains. We observed a modest, but significant, decrease of survival among larvae infected with *ume6*Δ compared to those infected with IV.1 in the first experiment (Fig. 8A), which was not reproduced in the second experiment (no significant difference, Fig. 8B). The comparison between *UME6*$^{HA}$ and IV.1 infections did not show any significant difference in terms of survival in both experiments (Fig. 8A and B). These results suggest that Ume6, despite its important role in *C. auris* morphogenesis, does not play a major role in virulence in the *Galleria* model.

### DISCUSSION

We investigated the role of the transcription factor Ume6 in *C. auris* by generating an hyperactivated Ume6 strain (*UME6*$^{HA}$) and an *UME6* deletion strain (*ume6*Δ). In *C. albicans*, Ume6 was shown to be a key regulator of the transition from the yeast to hyphal form and in biofilm formation, ultimately contributing to its virulence (28, 29, 42). We found a similar role of Ume6 in morphogenetic processes of *C. auris*, with also an implication in

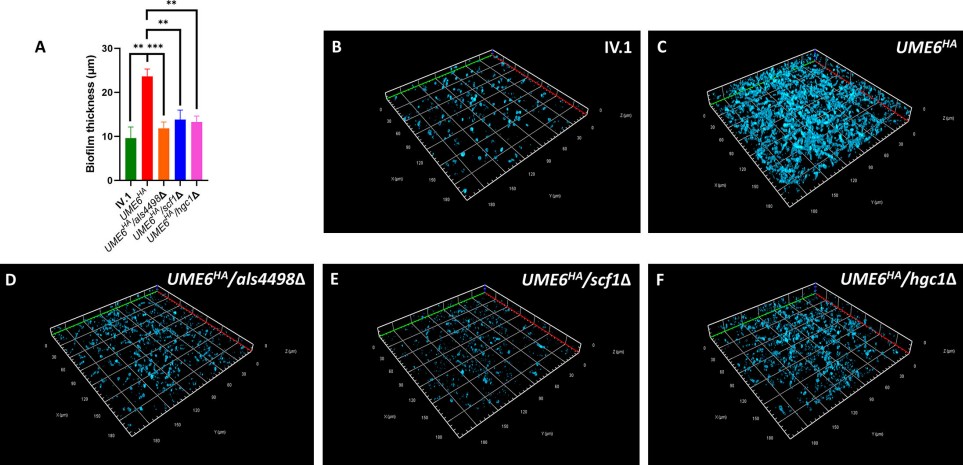

**FIG 7** Confocal microscopy. (A) Bar chart representing the biofilm thickness. Results are expressed as means and standard deviations (error bars) of three measurements taken on three different Z-stack pictures (*y*-axis) for the different strains (*x*-axis). Statistical analysis was performed using unpaired *t*-test with significant *P*-value defined as ≤0.05 (* ≤0.05, ** ≤0.01, *** ≤0.001, **** ≤0.0001). Representative Z-stack pictures taken by confocal microscopy of biofilms formed on glass coverslips with calcofluor white staining (at 630× magnification) are shown for the different strains: IV.1 (B), *UME6^HA* (C), *UME6^HA/als4498Δ* (D), *UME6^HA/scf1Δ* (E), and *UME6^HA/hgc1Δ* (F). Size standards are given for each axis. Minor ticks are at 6 µm scale.

aggregation, a unique feature of this emerging pathogen and some of its closely related species (e.g., *C. haemulonii*). Our further analyses (RNA-seq and selected gene deletions in the *UME6^HA* background) allowed us to decipher some important pathways involved in these processes, which are summarized in Fig. 9.

## Filamentation

Pseudohyphae or filaments resembling true hyphae have been observed in *C. auris* (19, 20, 43). Stress conditions inducing filamentation in *C. albicans* and *C. auris* are not the same. Filamentous forms of *C. auris* have been observed after a decrease in temperature, culture in high salt media, passage through a mammalian tissue, exposure to genotoxic substances (e.g., hydroxyurea, 5-fluorocytocine), disruption of the DNA damage-inducible non-coding RNA (DINOR), or Hsp90 inhibition or depletion (19, 20, 37, 43–45). While filamentation is an important virulence trait of *C. albicans*, its pathogenic role in *C. auris* is less obvious (20, 43, 46). Moreover, some genes expressed in *C. albicans* hyphal phase have no orthologs in *C. auris* (e.g., *ECE1*, *HWP1*) or no significant change of expression in different conditions inducing *C. auris* filamentation (e.g., *FLO8*, *EFG1*, *BRG1*, *BCR1*, *EFH1*), suggesting distinct pathways between these genotypically distant *Candida* spp. (20, 37, 47). Regarding *C. auris UME6*, significant overexpression of this gene was found in some *in vitro* filament-inducing conditions (e.g., hydroxyurea treatment) but not *in vivo* after passage through the mammalian body (20, 47). A comparison of our transcriptomic and GO enrichment analysis of *UME6^HA* with these previous data sets from other filament-inducing conditions in *C. auris* found some overlaps (Fig. 4; Fig. S3). For instance, genes involved in filamentous growth (*HGC1*, *DDR48*, *PHR1*) were upregulated in *UME6^HA* and in the filamentous phenotype of *C. auris* following passage through the mammalian body (20). *PHR1* was also upregulated following Hsp90 repression/inhibition, along with *SCF1* and other genes of cell surface proteins exhibiting increased expression in *UME6^HA* (*PGA26*, *RBT4/KRE1*) (37). The GSEA analysis presented here confirmed these overlaps with a significant higher number of genes commonly regulated by Ume6 and these transcriptional studies. While Kim et al. demonstrated a key role of the cyclic adenosine monophosphate/protein kinase A (cAMP/PKA) pathway in the filamentation of *C. auris* (47), we observed no relevant change of expression among genes involved in this pathway (*CYR1*, *BCY1*, *TPK1*, *TPK2*) following Ume6 hyperactivation (File S1).

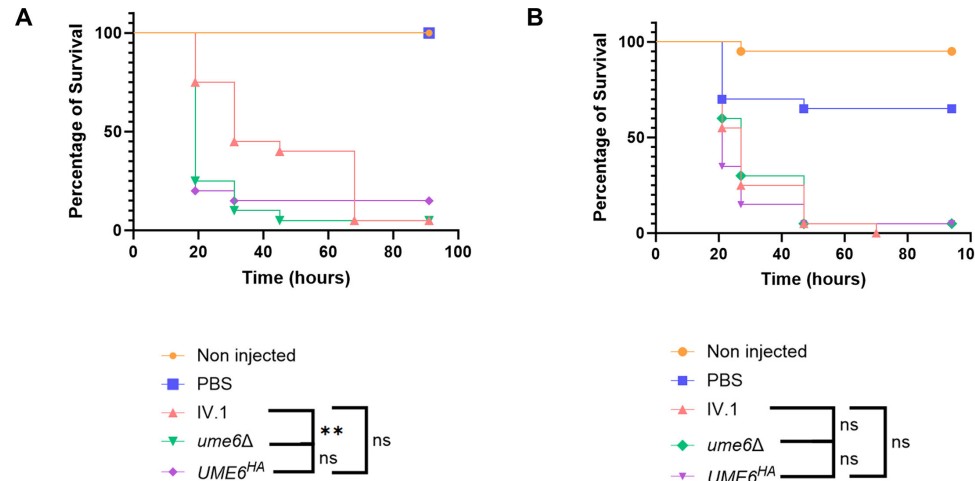

**FIG 8** Impact of *UME6* hyperactivation and deletion on virulence in a *Galleria mellonella* model of infection. Larvae were injected with the IV.1, *UME6^HA^*, or *ume6Δ* strains. Control groups consisted of non-injected larvae and larvae injected with PBS only. Graphs (A and B) represent two different experiments realized with 20 larvae per group. Statistical analyses were performed by log rank test with significant *P*-value defined as ≤0.05 (* ≤0.05, ** ≤0.01, *** ≤0.001, **** ≤0.0001).

We, therefore, conclude that mechanisms of filamentation in *C. auris* involve different pathways in response to different stress conditions. While Ume6 can induce filamentation via transcriptional activation of *HGC1*, its actual role in *C. auris* morphogenesis under real pathogenic conditions remains unclear.

## Aggregation

Aggregation was found to impact *C. auris* virulence. Non-aggregative variants were more virulent than aggregative variants in *Galleria* models (17, 24, 43, 46, 48). However, aggregation may favor skin colonization and possibly persistence in host tissues (49).

The ability of *C. auris* to form aggregates is variable among clinical isolates (18). Aggregation has been mainly observed among clade III isolates but has also been reported in other clades and can be induced under some *in vitro* or *in vivo* conditions (18, 49–51). Bing et al. showed that point mutations in genes involved in cell wall integrity (e.g., *CHS1*, *KIN3*, *ACE2*) or cell division (e.g., *KIC1*, *LAA1*) were associated with aggregation (50). Using a mutagenesis system, Santana et al. identified aggregating mutants with defects in daughter cell separation resulting from disruption of genes involved in cell wall homeostasis, such as chitinase regulators (*ACE2*, *TAO3*) or chitin synthase (*CHS2*) (52). Aggregation can result from adhesin-independent and adhesin-dependent mechanisms involving a defect in cell separation after budding and sticking of cells to each other, respectively (49). The adhesin-independent mechanism can be induced in all clades by exposure to some antifungal drugs (echinocandins, azoles) (18, 49). Conversely, the adhesin-dependent mechanism has been mainly observed in clade III isolates (18, 49). In clade III, the adhesin Als4112 (SBP28_005090/B9J08_004112) was shown to play a key role in aggregation via copy number variation (24, 49). Deletion of *ALS4112* in an aggregative clade I isolate resulted in aggregation defect (40). In our transcriptomic analysis, *ALS4112* exhibited a modest increase of expression (twofold) following Ume6 hyperactivation (File S1). We found here that Ume6 triggered aggregation mainly via two other adhesins, Als4498 and Scf1. Interestingly, we did not find increased expression of *UME6* in an aggregative clade III isolate (compared to the non-aggregative IV.1 strain, data not shown). We, therefore, conclude that the Ume6-dependent pathway of aggregation described here may be distinct from that observed among clade III isolates. Further analyses would be warranted to assess the role of Ume6 in aggregative clinical isolates and the impact of *UME6* overexpression or deletion in other clades.

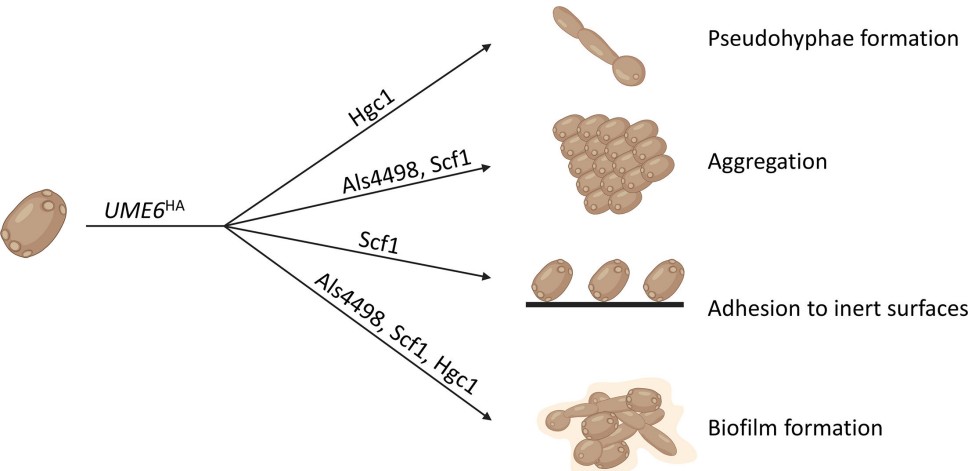

**FIG 9** Schematic representation of the Ume6-dependent pathways controlling aggregation, pseudohyphae formation, adhesion, and biofilm formation. The transcription factor Ume6 controls important morphogenetic processes of *C. auris* via different modulators: (i) pseudohyphae formation via Hgc1, (ii) aggregation via Als4498 and Scf1, (iii) adhesion to inert surfaces via Scf1, and (iv) biofilm formation via Als4498, Scf1, and Hgc1.

## Adhesion/ biofilm formation

*C. auris* displays at least three Als-like, eight Hyr/Iff-like, and one species-specific adhesin (Scf1) (27, 53–55) (Table S3). Some of these adhesins were shown to be upregulated in biofilm conditions (26). Following Ume6 hyperactivation, we observed a drastic overexpression of *ALS4498* and *SCF1* (>1,000-fold) and, to a lesser extent, of *HYR3* and *IFF9* (10- to 100-fold), which suggests that these genes can be under the direct control of Ume6. In *C. albicans*, Als1 was found to be important for adhesion (56). Scf1 is unique to *C. auris* and has a different adhesion mechanism compared to other adhesins (27). The crucial role of Scf1 in adhesion and biofilm formation has been recently demonstrated (27, 40). Our results showed that adhesion to inert surfaces in *UME6*^HA was essentially mediated by Scf1. Interestingly, we found that Als4498 had a predominant role in biofilm formation, but not in adhesion to inert surfaces. In addition, both Scf1 and Hgc1 were found to play a role in biofilm formation of *UME6*^HA. Biofilm formation is a complex process. While adhesion to living or inert surfaces represents the first stage, other steps are required including morphogenetic changes, proliferation, and maturation (14). *C. auris* biofilms are distinct from that of *C. albicans* and usually consist of yeast cells with a low amount of extracellular matrix (12). Aggregation seems to play an important role in *C. auris* biofilm formation, since aggregating strains demonstrated higher biofilm-forming capacity, which was associated with distinct transcriptomic profiles compared to the non-aggregative strains (13, 23, 25, 57). Therefore, we postulate that Als4498 is important in biofilm formation by promoting intercellular adhesion and aggregation rather than adhesion to the inert surface, as this later mainly relies on Scf1. Hgc1 also contributes to biofilm formation, possibly by promoting filamentation at the proliferation stage. Finally, the thick biofilm resulting from Ume6 hyperactivation was more resistant to amphotericin B and micafungin (two antifungal drugs with anti-biofilm activity), which might have clinical implications for the treatment of catheter-related infections.

## Conclusions

The present work demonstrated the role of the transcription factor Ume6 in *C. auris* morphogenesis with distinct pathways in filamentation, aggregation, adhesion, and biofilm formation (Fig. 9). We also highlight a link via Ume6 between different morphogenetic processes, such as aggregation and filamentation. Indeed, Garcia-Bustos et al. have observed a higher propensity to filamentation in aggregative phenotypes (46).

In addition to these Ume6-dependent pathways, Ume6-independent pathways may operate in these processes and their respective roles remain to be deciphered in clinical isolates. Of note, there is currently no demonstration of gain of function mutations in Ume6 in available isolate collections, which could result in its hyperactivation similar to our artificial hyperactivation model. Moreover, while Ume6 was shown to play a role in *C. albicans* virulence (28, 42), we could not reproduce this impact in an insect model of *C. auris* infection. This is in line with previous observations showing that filamentation of *C. auris* has a limited impact on virulence, while aggregation has been associated with decreased virulence (17, 46). However, the role of Ume6 on morphogenetic plasticity may have important consequences for pathogenicity, for instance in interhuman transmission, biofilm formation, or antifungal resistance of biofilms.

## MATERIALS AND METHODS

### Plasmids and strains

Plasmids pDS2020 containing the *NatR* cassette (nourseothricin resistance) and pYM70 containing the *HygR* cassette (hygromycin resistance) were used for the construction of the deletion strains (32). Plasmid pjli8, constructed from plasmid Clp-p*ACT1*-3xFLAG-MNase-SV40-*CYC-SAT1* containing the nourseothricin resistance cassette *SAT1* and the *C. auris* neutral site *CauNI* (32, 58), was used for the construction of the hyperactivated Ume6 strain (*UME6^HA*). *Escherichia coli* DH5α was used for plasmid generation, as previously described (32). Plasmids were extracted with the Plasmid Mini Kit (Qiagen, Hilden, Germany). Primers used in this study are listed in Table S2. The *C. auris* isolate IV.1 [clade IV, LMDM 1219 (59)] was used as a source for DNA amplification and as a background strain for genetic transformations (32). Yeast extract-peptone-dextrose (YEPD) containing bactopeptone 20 g/L, yeast extract 10 g/L, glucose 20 g/L with or without agar 20 g/L was used as culture medium. All cultures were incubated at 37°C on solid YEPD agar plates or in liquid YEPD under constant agitation (220 rpm).

### Genetic transformations

For the construction of the *UME6^HA* strain, the *UME6* gene was cloned at KasI/BsrGI sites of plasmid pjli8 under the control of the *ADH1* promoter with a 3×HA tag at its C-terminal locus, as previously described (32) (Fig. S7). The constructs for deletion strains were obtained by fusion PCR, as previously described (32). The selection cassette (*NatR* or *HygR*) was flanked by sequences of approximately 500 bp of the upstream and downstream regions of the target gene (Fig. S8 to S11).

Transformation in *C. auris* IV.1 was performed by CRISPR-Cas9 and by the electroporation protocol, as previously described (32). Specific RNA-guides were designed to contain 20 bp homologous sequences of the upstream and downstream regions of the target region (Table S2). Transformants were selected at 37°C on YEPD containing 200 µg/mL of nourseothricin (Werner BioAgents, Jena, Germany) or 600 µg/mL of hygromycin B (Corning, Corning, NY) according to the selection marker. Integration of the constructs was verified by PCR (Fig. S12 to S25).

HA tagging of the Ume6 protein was verified by Western blot using a HA tag monoclonal antibody (Invitrogen, ThermoFisher Scientific, Waltham, MA), as previously described (Fig. S26) (32).

### Transcriptomic analyses

The strains were grown overnight in liquid YEPD. Concentrations were adjusted to an optical density (OD) corresponding to approximately $0.75 \times 10^7$ cells/mL with an additional 3 h incubation to reach approximately $1.5 \times 10^7$ cells/mL. Samples were prepared in triplicates for each strain. RNA was extracted with Quick-RNA fungal/bacterial miniprep kit (Zymo Research, Freiburg im Brisgau, Germany). RNA extracts were treated with the Turbo DNA-free kit (Thermo Fisher Scientific Inc., Waltham, MA).

RNA concentration was measured with NanoDrop 1000 spectrophotometer (Witec AG, Switzerland) and adjusted to a concentration of 9 ng/µL in RNA-free water. RNA quality was assessed on Fragment Analyzer (Agilent Technologies, Santa Clara, CA) with RNA quality numbers between 8.3 and 8.6. RNA-seq libraries were prepared from 250 ng of total RNA with the Illumina Stranded mRNA Prep reagents (Illumina, San Diego, CA). Libraries were quantified by a fluorometric method (QubIT, Life Technologies, Carlsbad, CA) and their quality assessed on a Fragment Analyzer (Agilent Technologies). Sequencing was performed on an Illumina NovaSeq 6000. Sequencing data were demultiplexed using the bcl2fastq2 Conversion Software (version 2.20, Illumina). Reads were aligned to the *C. auris* genome (isolate VI.1, Bioproject PRJNA1036037) using a RNAseq analysis workflow with CLC genomic Workbench (Version 23). Data are available in the bioproject PRJNA1036037.

## Gene ontology term enrichment analysis

The gene ontology (GO) term analysis was performed with genes selected by *P* values of ≤ 0.05 and log2 fold-change ≥1 (upregulated genes) or ≤−1 (downregulated genes). Gene orthologs (B8441) to the *C. auris* isolate IV.1 were established by systematic local blast using OmicsBox (3.1.9, BioBam Bioinformatics) of the IV.1 orfeome with B8441 orfeome data available from NCBI (GCA_002759435.2). The gene lists of B8441 orthologs (File S1) were used in FungiDB and implemented in the GO term analysis tool (60).

## Gene set enrichment analysis

GSEA was produced from data in File S1 ("Ume6tag up genes," "Ume6tag down genes," "Cauris.gmt"), in which Ume6-regulated genes ("Ume6.rnk") with *P* values of ≤ 0.05 and log fold-change ≥1 or ≤−1 were chosen. The gene list ("Cauris.gmt") contains 90 differential expression data sets (up- and downregulated genes) from published transcriptional data performed with *C. auris* (references in Cauris.gmt file), which was imported into the GSEA software (4.3.2). Analysis parameters were as follows: norm, meandiv; scoring_scheme, weighted; set_min, 5; nperm, 1000; set_max, 1000. GSEA results were uploaded into Cytoscape 3.8.2 with the following parameters: *P* value cutoff, 0.01; FDR *q* value, 0.05.

## Crystal violet assay for biofilm quantification

The strains were cultured overnight in liquid YEPD, washed with PBS, and resuspended in Roswell Park Memorial Institute (RPMI) medium at a density of approximately $1 \times 10^6$ cells/mL. Suspensions of 200 µL were incubated in a flat-bottomed polystyrene untreated Costar 96-well plate (Corning Inc., Corning, NY) at 37°C without agitation for 24 h. The wells were washed with PBS and 100 µL of crystal violet 0.5% was added to each well. After 5 min incubation at room temperature, the wells were washed with distilled water and 200 µL of ethanol 95% was added. The samples were then transferred in clean wells, and the absorbance (590 nm) was analyzed at 24 h. The experiment was performed in technical triplicates and biological duplicates for each strain. Mean absorbances (representing the biomass of the biofilm) of the different conditions (strains) were compared using the unpaired *t*-test.

## Flow cytometry

Adhesion to fluorescent polystyrene microspheres as a readout for adhesion to hydrophobic surfaces can be analyzed by flow cytometry as described previously (61). In this protocol, microspheres act as a surface to which yeast cells can adhere enabling their quantification. The strains were grown overnight in liquid YEPD. Yeast suspensions were adjusted to $1.5 \times 10^7$ cells/mL and incubated with fluorescent microspheres (FluoSpheres Carboxylate-Modified Microspheres, ThermoScientific, Waltham, MA) for 1 h at 25 rpm in aluminum foil. They were then fixed with 2× volume of ethanol 100% and incubated overnight at 4°C in the dark. The samples were centrifuged at 6,000 rpm for 5 min, and

the pellet was rehydrated for 30 min in 50 mM Na-citrate buffer (pH 7). Permeabilization was achieved by incubation with RNAse A (0.25 mg/mL) at 55°C for 1 h and then with proteinase K (20 mg/mL) for 1 h. Cells were stained with Vybrant DyeCycle Green Stain (ThermoFisher Scientific Waltham, MA). Flow cytometric analysis was performed using a CytoFlex LX Flow Cytometer (Beckman Coulter Diagnostics, Brea, CA), and data were analyzed by FlowJo software, version 10.10.0 (BD Biosciences, San Jose, CA). Cells were delineated based on the uptake of DyeCycle Green Stain, which identifies nucleated cells, characterized by their emission within the green fluorescence spectrum ($\lambda_{em} \approx 488$ nm). Following initial gating on DyeCycle Green-positive events, cells were further analyzed for adherence to fluorescent microspheres characterized by their emission within the red fluorescence spectrum ($\lambda_{em} \approx 645$ nm) indicating bead binding. The mean proportions of adherent cells of different conditions (strains) were compared using unpaired $t$-test.

Imaging flow cytometry was performed to analyze different morphological aspects (62). Samples were acquired using a 5-laser 12 channel ImageStreamX imaging flow cytometer (Cytek Biosciences, Fremont, CA) at low speed and highest magnification (60×). Cells were excited with a 488 nm laser (12 mW), a 642 nm laser (1.5 mW), and a 785 nm Side Scatter (SSC) laser (1.5 mW). Data were acquired for at least 25,000 events/sample. Experimental samples contained images for bright-field (430–480 nm and 560–595 nm), DyeCycle Green (505–560 nm), FluoSpheres microspheres (642–745 nm), and Side Scatter (745–800 nm). Events with a bright-field area >1 $\mu m^2$ (to exclude cell debris) and non-saturating pixels (Raw max pixel values below 4,096) were collected. Data analysis was done using Image Data Exploration and Analysis Software (IDEAS) version 6.3 (Cytek Biosciences, Fremont, CA). Single color controls for each fluorochrome were acquired to generate the compensation matrix that was applied to each sample prior to analysis (63). The gating strategy shown in Fig. S27 was used to separate the cells of interest from the out-of-focus cells, debris, and clumps, as previously reported (64). The adhesion of yeast cells to microspheres was quantified by plotting the intensities of DyeCycle Green and FluoSpheres microspheres. Cell morphology was analyzed based on nuclear size (DyeCycle Green area) and shape (major/minor axis ratio).

## Confocal microscopy

Yeasts were grown overnight in liquid YEPD. A glass-coverslip (VWR, diameter 12 mm) was placed in each well of a 24-wells plate (Corning Inc., Corning, NY) and 0.01 mg/mL poly-D-lysine (Sigma-Aldrich, St Louis, MO) was added for 30 min. After three washes with milliQ water, the plate was dried under the hood. Yeasts were washed in PBS, and the density was adjusted to $1 \times 10^7$ cells/mL in RPMI. This suspension was added in the wells and incubated for 90 min at 37°C under 75 rpm. The wells were then washed with PBS, and fresh RPMI was added on the glass-coverslips. After 48 h incubation, the wells were washed with PBS and formaldehyde 4% was added for 30 min. The samples were washed again and stained with 0.1 mg/mL Calcofluor White (Sigma-Aldrich, St Louis, MO) for 15 min. Analysis was performed with a confocal microscope (Zeiss LSM900) using an immersion oil 63× objective. Z-stack pictures were taken with the software ZEN 3.2, with a Z-interval of 1 $\mu m$. The mean thicknesses of the biofilm of the different conditions (strains) were compared using the unpaired $t$-test.

## Scanning electron microscopy

SEM was performed as previously described (65) with some adaptations. Falcon 24-well Clear Multiwell Plate (Corning Inc., Corning, NY) were used. NUNC Thermanox coverslips (ThermoFisher Scientific, Waltham, MA) were coated in 0.01 mg/mL poly-D-lysine for 30 min and washed in sterile Milli-Q water. The coverslips were dried under a hood before adding a drop of yeast suspension adjusted to a density of $1.5 \times 10^7$ cells/mL. The coverslips were washed briefly with PBS, fixed with glutaraldehyde 1.5% (Electron Microscopy Sciences, Hatfield, PA) for 30 min, and washed again. Then, samples were post-fixed with 1% osmium tetroxide and 1.5% potassium hexacyanoferrate(II) trihydrate for 1 h and then rinsed in water. Dehydration was performed in ethanol solutions (30%,

50%, and 70% for 40 min each, and 100% for 1 h). Samples were then processed in a Critical Point Dryer (Leica, Wetzlar, Germany) and coated in a sputter coater (Safematic GmbH, Zizers, Switzerland) with 10 nm gold palladium. Coverslips were mounted on aluminum stubs with carbon tape and observed with a Quanta 250 FEG scanning electron microscope (ThermoFisher Scientific, Waltham, MA) at 10 kV (spot size 4.5, working distance of approximately 8.5 mm) using the in-chamber secondary electron detectors (ETD) and in-chamber backscattered electron detector (BSED) at several magnifications (2,400× and 8,000×).

## Antifungal susceptibility testing

Antifungal susceptibility to fluconazole, amphotericin B (Sigma-Aldrich, St Louis, MO), and micafungin (Selleck Chemicals, Houston, TX) was tested in planktonic and biofilm conditions. MICs in planktonic conditions were determined according to the CLSI protocol (41). MBECs in biofilm conditions were assessed by the tetrazolium salt (XTT) reduction assay as previously described (66, 67), with minor modifications. After overnight growth, yeast suspensions were adjusted to $10^7$ cells/mL in RPMI medium. Then, 100 µL of this suspension was added in wells of a 96-wells plate (Corning Inc., Corning, NY) and incubated at 37°C for 90 min at 75 rpm. The wells were washed with PBS, and 200 µL of fresh RPMI was added. After 24 h incubation (37°C, 75 rpm), 200 µL of antifungal drug was added in each well with a gradient of concentration. After 48 h of incubation, the wells were washed and 200 µL of a PBS solution containing XTT (1 mg/mL) and menadione (0.4 mM) was added. The plate was incubated for 2.5 h at 37°C in the dark without shaking and absorbance was read at 490 nm. The minimum biofilm eradication concentration achieving 50% inhibition ($MBEC_{50}$) of metabolic activity compared to the positive control well was assessed for each drug. Testing was performed in biological duplicates and validated if results were similar (± one dilution).

## Virulence assay with *Galleria mellonella*

*Galleria mellonella* larvae (Bait Express GmbH, Basel, Switzerland) weighting 425–525 mg were stored at 10°C before the experiment. The *C. auris* strains were grown overnight in liquid YEPD. Cultures were washed with PBS and resuspended at a concentration of approximately $1.25 \times 10^7$ cells/mL. An inoculum of 40 µL of the yeast suspension was injected in the larvae with insulin syringes (Micro-Fine, BD, Franklin Lakes, NJ) after disinfection with ethanol 70%. Larvae were incubated in Petri dishes with clean sawdust in darkness at 37°C. Survival was monitored twice a day for 4 days. Kaplan-Meier curves of survival were designed with Graphpad Prism 9.0 software and statistical analyses were performed by the log-rank test.

## ACKNOWLEDGMENTS

We are grateful to Hannes Richter, Julien Marquis, and Johann Weber, from the Genomic Technologies Facility of the University of Lausanne (Lausanne, Switzerland), for processing of transcriptomic samples and data.

This study was supported by grants from the Swiss National Science Foundation (SNSF, project number 310030_192611), by the Santos-Suarez foundation, and by the Carigest SA foundation.

Lamoth has received research grants from Novartis, MSD, Gilead, and Pfizer outside of the present work, and speaker honoraria from MSD, Gilead, Pfizer, Mundipharma, and Becton-Dickinson. All fees were paid to his institution (CHUV).

## AUTHOR AFFILIATIONS

[1]Institute of Microbiology, Lausanne University Hospital and University of Lausanne, Lausanne, Switzerland
[2]Flow Cytometry Facility, University of Lausanne, Lausanne, Switzerland

[3]Electron Microscopy Facility, University of Lausanne, Lausanne, Switzerland
[4]Infectious Diseases Service, Department of Medicine, Lausanne University Hospital and University of Lausanne, Lausanne, Switzerland

## AUTHOR ORCIDs

Nicolas Jacquier  http://orcid.org/0000-0002-1974-8161
Dominique Sanglard  http://orcid.org/0000-0002-5244-4178
Frederic Lamoth  http://orcid.org/0000-0002-1023-5597

## FUNDING

| Funder | Grant(s) | Author(s) |
| --- | --- | --- |
| Swiss National Science Foundation (SNSF) | 310030_192611 | Frederic Lamoth |
| Fondation Santos-Suarez pour la Recherche Médicale (Santos-Suarez Foundation for Medical Research) | | Frederic Lamoth |
| Carigest (Carigest SA) | | Frederic Lamoth |

## AUTHOR CONTRIBUTIONS

Marine Louvet, Conceptualization, Data curation, Formal analysis, Investigation, Methodology, Writing – original draft | Jizhou Li, Conceptualization, Data curation, Formal analysis, Investigation, Methodology, Writing – original draft | Danielle Brandalise, Data curation, Formal analysis, Investigation, Methodology, Writing – review and editing | Daniel Bachmann, Data curation, Formal analysis, Investigation, Methodology, Writing – review and editing | Francisco Sala de Oyanguren, Data curation, Formal analysis, Investigation, Methodology, Writing – review and editing | Danny Labes, Data curation, Formal analysis, Investigation, Methodology, Writing – review and editing | Nicolas Jacquier, Data curation, Formal analysis, Investigation, Methodology, Writing – review and editing | Christel Genoud, Data curation, Formal analysis, Investigation, Methodology, Writing – review and editing | Antonio Mucciolo, Data curation, Formal analysis, Investigation, Methodology, Writing – review and editing | Alix T. Coste, Conceptualization, Formal analysis, Investigation, Methodology, Writing – review and editing | Dominique Sanglard, Conceptualization, Formal analysis, Investigation, Methodology, Software, Supervision, Writing – review and editing | Frederic Lamoth, Conceptualization, Formal analysis, Funding acquisition, Investigation, Methodology, Project administration, Supervision, Writing – original draft, Writing – review and editing

## ADDITIONAL FILES

The following material is available online.

### Supplemental Material

**Supplemental figures (Spectrum01531-24-s0001.pdf).** Fig. S1 to S27.
**Supplementary File S1 (Spectrum01531-24-s0002.xlsx).** Comparative transcriptomic analysis of the strains UME6HA and IV.1.
**Supplementary File S2 (Spectrum01531-24-s0003.xlsx).** GO terms of molecular functions for upregulated genes.
**Supplementary File S3 (Spectrum01531-24-s0004.xlsx).** Detailed list of genes (n=46).
**Supplemental tables (Spectrum01531-24-s0005.docx).** Tables S1 to S3.

### Open Peer Review

**PEER REVIEW HISTORY (review-history.pdf).** An accounting of the reviewer comments and feedback.

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
