## [Reviewer comments · Microbiology Spectrum]

Microbiology Spectrum

Ume6-dependent pathways of morphogenesis and biofilm formation in *Candida auris*

Marine Louvet, Jizhou Li, Danielle Brandalise, Daniel Bachmann, Francisco Sala de Oyanguren, Danny Labes, Nicolas Jacquier, Christel Genoud, Antonio Mucciolo, Alix Coste, Dominique Sanglard, and Frederic Lamothe

Corresponding Author(s): Frederic Lamothe, Centre Hospitalier Universitaire Vaudois

Review Timeline:

Submission Date:	June 27, 2024
Editorial Decision:	July 19, 2024
Revision Received:	August 2, 2024
Accepted:	August 5, 2024

Editor: Alexandre Alanio

Reviewer(s): Disclosure of reviewer identity is with reference to reviewer comments included in decision letter(s). The following individuals involved in review of your submission have agreed to reveal their identity: Alexander Lorenz (Reviewer #2)

Transaction Report:

DOI: <https://doi.org/10.1128/spectrum.01531-24>

Re: Spectrum01531-24 (Ume6-dependent pathways of morphogenesis and biofilm formation in *Candida auris*)

Dear Dr. Frederic Lamoth:

Thank you for the privilege of reviewing your work. Below you will find my comments, instructions from the Spectrum editorial office, and the reviewer comments.

Revision Guidelines

Sincerely,
Alexandre Alanio
Editor
Microbiology Spectrum

Reviewer #1 (Comments for the Author):

The article by Louvet et al describes the role of transcription factor Ume6 in regulating morphogenesis and biofilm formation in *Candida auris*. *C. auris* is a multi-drug resistant pathogen well-known for causing nosocomial infections. The article presented is well written and supported by strong datasets. Here are my comments:

1) Does the deletion of the gene Ume6 has any effect on the morphogenesis and biofilm formation? In other words can

targeting Ume6 may affect *C. auris* pathogenicity? Please explain

2) How much fold increase was the expression of hyperactive Ume6 was observed compared to the wild-type. Please explain.

3) Lines 338 and 339, the numbers should be 0.75 or 1.5 and not 0,75 and 1,5.

4) Fig 8A, the survival rates between WT and both mutants are almost two fold, where as in Fig 8B, it seems no difference were observed. Is the change significant in Fig 8A? Did the authors perform any statistical tests to see whether the change is significant or not? Please explain.

Reviewer #2 (Comments for the Author):

This manuscript by Louvet et al. reports how the zinc-cluster transcription factor Ume6 influences cellular morphogenesis in the opportunistic fungal pathogen *Candidozyma auris* (please note that this organism has recently been renamed, <https://doi.org/10.3767/persoonia.2024.52.02>). The authors generated a hyperactive version of Ume6 and also a deletion of the UME6 gene. They established using transcriptomics, fluorescence & electron microscopy, flow cytometry, and genetics that Ume6 affects biofilm formation, aggregation, and filamentation via different pathways.

The data is of good quality and supports the authors' conclusions. Overall, I found the manuscript an easy and enjoyable read and the story convincing. I have a few comments to clarify some points and hopefully help to improve the article further.

Specific Comments:

Personally, I think it's time to stop calling *Candidozyma auris* "an emerging pathogen" (l.4, l.34), it has emerged 15 years ago and has since established itself as an agent of nosocomial disease.

The assertion that Ume6 was hyperactivated by a C-terminal HA-tag is very confusing (l.10-11, l.71-73) until it is properly explained (l.73-75). I would remove "by C-terminal tagging ... (HA)" in the abstract and reword l.71-75 to make this more accessible.

Please follow correct nomenclature: pADH1 (l.72) indicates a plasmid not a promoter. A promoter would be a capital P followed by the gene name in italics and subscript.

l.102 better "locate to the cell wall"

I find it commendable that the authors adopted the ALS gene naming convention put forward by Santana et al. (their ref.26), this didn't originate from Pelletier et al. (their ref.37) as they mistakenly state. However, they ignore the reasoning of this naming convention, namely that orthology of ALS factors cannot be assigned between species. The phylogenetic analysis of ALS genes indicates that all ALS genes within a given species cluster with each other and not with equivalent ALS genes from other species (<https://doi.org/10.1093/genetics/iyab029>), i.e. Als4498 is not the ortholog of *Candida albicans* Als1 (l.118-119), nor is Als4112 the ortholog of *Candida albicans* Als4 (l.249) or the ortholog of Als3 or Als5 as has been suggested by other authors.

The principle of the adhesion assay (polystyrene beads analysed by flow cytometry) needs to be explained (l.154), currently unclear what this measures and how.

Figures 5 and 6 could be merged to reduce an overall rather high number of figures.

l.315: Please specify the type of peptone (bacto- or mycopeptone?) and give full recipe of the medium.

Error bars in figures 5, 6, and 7A have not been defined. Use of unpaired T-test in these figures has not been justified. Please note that that figure 7A shows a bar chart not a histogram as stated in the legend.

Reviewer #1 (Comments for the Author):

The article by Louvet et al describes the role of transcription factor Ume6 in regulating morphogenesis and biofilm formation in *Candida auris*. *C. auris* is a multi-drug resistant pathogen well-known for causing nosocomial infections. The article presented is well written and supported by strong datasets. Here are my comments:

1) Does the deletion of the gene *Ume6* has any effect on the morphogenesis and biofilm formation? In other words can targeting *Ume6* may affect *C. auris* pathogenicity? Please explain

Response: Deletion of *UME6* had no impact on morphogenesis (pseudo-filamentation and aggregation), adhesion and biofilm formation. We agree that these results are partially lacking. We have now added a supplementary figure (new Figure S1) to show the results of phenotypic characteristics (panel A), adhesion (panel B) and biofilm formation (panel C) of the *ume6Δ* strain. The manuscript has also been modified accordingly (lines 78, 155-156 and 163-164).

2) How much fold increase was the expression of hyperactive *Ume6* was observed compared to the wild-type. Please explain.

Response: Our transcriptomic analysis showed that *UME6* exhibited a 2210-fold overexpression in the *UME6^{HA}* strain (i.e. hyperactive *Ume6*) compared to the IV.1 strain (background wild-type strain), which is stated in the manuscript (lines 98-99). Raw data of this transcriptomic analysis is also displayed in Supplementary File S1.

3) Lines 338 and 339, the numbers should be 0.75 or 1.5 and not 0,75 and 1,5.

Response: Thank you, this has been corrected.

4) Fig 8A, the survival rates between WT and both mutants are almost two fold, where as in Fig 8B, it seems no difference were observed. Is the change significant in Fig 8A? Did the authors perform any statistical tests to see whether the change is significant or not? Please explain.

Response: We have now added statistical tests in Figure 8 (see revised figure). Indeed, we have repeated the statistical comparisons and found that the difference of survival curves between *ume6Δ* and IV.1 was significant (albeit modest) in the first experiment (panel A). However, it was not reproducible (not significant) in the second experiment (panel B). We have added a comment about this in the results section (lines 189-192).

Reviewer #2 (Comments for the Author):

This manuscript by Louvet et al. reports how the zinc-cluster transcription factor *Ume6* influences cellular morphogenesis in the opportunistic fungal pathogen *Candidozyma auris* (please note that this organism has recently been renamed, <https://doi.org/10.3767/persoonia.2024.52.02>). The authors generated a hyperactive version of *Ume6* and also a deletion of the *UME6* gene. They established using transcriptomics, fluorescence & electron microscopy, flow cytometry, and genetics that *Ume6* affects biofilm formation, aggregation, and filamentation via different pathways. The data is of good quality and supports the authors' conclusions. Overall, I found the manuscript an easy and enjoyable read and the story convincing. I have a few comments to clarify some points and hopefully help to improve the article further.

Specific Comments:

Personally, I think it's time to stop calling *Candidozyma auris* "an emerging pathogen" (l.4,

I.34), it has emerged 15 years ago and has since established itself as an agent of nosocomial disease.

Response: We have now mentioned the new name “Candidozyma auris” in the introduction (line 34) with appropriate reference (ref 2) in the introduction. We have also suppressed the word “emerging” in the abstract and introduction.

The assertion that Ume6 was hyperactivated by a C-terminal HA-tag is very confusing (I.10-11, I.71-73) until it is properly explained (I.73-75). I would remove "by C-terminal tagging ... (HA)" in the abstract and reword I.71-75 to make this more accessible.

Response: We have removed the mention of the C-terminal HA tag in the abstract and we have reworded the paragraph explaining the principle of HA-tagging (lines 72-77) as suggested.

Please follow correct nomenclature: pADH1 (I.72) indicates a plasmid not a promoter. A promoter would be a capital P followed by the gene name in italics and subscript.

Response: Thank you, this has been corrected.

I.102 better "locate to the cell wall"

Response: Thank you, this has been corrected.

I find it commendable that the authors adopted the ALS gene naming convention put forward by Santana et al. (their ref.26), this didn't originate from Pelletier et al. (their ref.37) as they mistakenly state. However, they ignore the reasoning of this naming convention, namely that orthology of ALS factors cannot be assigned between species. The phylogenetic analysis of ALS genes indicates that all ALS genes within a given species cluster with each other and not with equivalent ALS genes from other species (<https://doi.org/10.1093/genetics/iyab029>), i.e. Als4498 is not the ortholog of *Candida albicans* Als1 (I.118-119), nor is Als4112 the ortholog of *Candida albicans* Als4 (I.249) or the ortholog of Als3 or Als5 as has been suggested by other authors.

Response: Thank you for these precisions. We have referred the nomenclature to Santana et al. as suggested (lines 118-120, ref 27). The genes are called *ALS4498* and *ALS4112*. Attempts to assign *C. albicans* orthology have been removed.

The principle of the adhesion assay (polystyrene beads analysed by flow cytometry) needs to be explained (I.154), currently unclear what this measures and how.

Response: We have added some introduction about the principle of this approach (lines 396-397) with appropriate reference (ref 61). We have also added a paragraph with more detailed explanation about the technical procedure for measurements (lines 409-415).

Figures 5 and 6 could be merged to reduce an overall rather high number of figures.

Response: We agree that the number of figures could be reduced. However, Figures 5 and 6 show distinct results (i.e. adhesion and biofilm formation) resulting from completely distinct analytical procedures. Therefore, to avoid any confusion, we preferred to keep these figures separated.

I.315: Please specify the type of peptone (bacto- or mycopeptone?) and give full recipe of the medium.

Response: We have added the detailed recipe of YEPD (lines 322-324).

Error bars in figures 5, 6, and 7A have not been defined. Use of unpaired T-test in these figures has not been justified. Please note that that figure 7A shows a bar chart not a histogram as stated in the legend.

Response: Error bars represent standard deviations, this has now been clearly stated in figure legend (lines 811-812, 823-824, 829-830). Unpaired t-test was used for

comparison of mean measurements between different conditions (strains), this has now been stated (lines 391-393, 414-415 and 445-447). The term “histogram” was substituted by “bar chart” as requested (line 814).

Re: Spectrum01531-24R1 (Ume6-dependent pathways of morphogenesis and biofilm formation in *Candida auris*)

Dear Dr. Frederic Lamoth:

Your manuscript has been accepted, and I am forwarding it to the ASM production staff for publication. Your paper will first be checked to make sure all elements meet the technical requirements. ASM staff will contact you if anything needs to be revised before copyediting and production can begin. Otherwise, you will be notified when your proofs are ready to be viewed.

Sincerely,
Alexandre Alanio
Editor
Microbiology Spectrum